# Predictors of Motivation to Receive a COVID-19 Vaccination and the Number of COVID-19 Vaccine Doses Received in Patients with Schizophrenia

**DOI:** 10.3390/vaccines11121781

**Published:** 2023-11-29

**Authors:** Chien-Wen Lin, Yu-Ping Chang, Cheng-Fang Yen

**Affiliations:** 1Department of Psychiatry, Kaohsiung Medical University Hospital, Kaohsiung 80708, Taiwan; 2Department of Psychiatry, School of Medicine, Kaohsiung Medical University, Kaohsiung 80708, Taiwan; 3School of Nursing, The State University of New York, University at Buffalo, New York, NY 14214-8013, USA; 4College of Professional Studies, National Pingtung University of Science and Technology, Pingtung 91201, Taiwan

**Keywords:** COVID-19, schizophrenia, stigma, vaccine

## Abstract

Individuals with schizophrenia are more likely to be infected with COVID-19 than are members of the general population. No prospective study has examined the associations of multi-dimensional factors with the motivation to receive vaccination against COVID-19. This follow-up study investigated the effects of individual (sociodemographic and illness characteristics, depression, and self-esteem), environmental (perceived social support), and individual–environmental interaction factors (self-stigma and loneliness) on the motivation to receive vaccination against COVID-19 and the number of COVID-19 vaccine doses received one year later among 300 individuals with schizophrenia. The associations of baseline factors with motivation to receive vaccination against COVID-19 and the number of vaccine doses received 1 year later were examined through linear regression analysis. The results indicated that greater loneliness (*p* < 0.01) and being married or cohabitating (*p* < 0.05) at baseline were significantly associated with lower motivation to receive vaccination against COVID-19 at follow-up. Disorganization (*p* < 0.05) at baseline was significantly associated with fewer COVID-19 vaccine doses at follow-up; greater motivation to receive vaccination was significantly associated with more COVID-19 vaccine doses at follow-up (*p* < 0.001). Health professionals should consider the identified predictors while developing intervention programs aimed at enhancing vaccination against COVID-19 in individuals with schizophrenia.

## 1. Introduction

Since the end of 2019, the COVID-19 pandemic has affected the world. On 2 November 2023, the World Health Organization (WHO) reported over 771,679,618 confirmed COVID-19 cases, including 6,977,023 deaths, worldwide [1]. Vaccines are among the most vital measures for preventing COVID-19 [2]. However, hesitancy to receive the COVID-19 vaccine is prevalent. Vaccine hesitancy is defined by the WHO as a “delay in acceptance or refusal of vaccines despite availability of vaccination services” [3]. A meta-analysis reported that a quarter of the population is hesitant to receive the COVID-19 vaccine [4]. Understanding the factors influencing individuals’ decisions regarding vaccination is the first step for developing strategies to enhance individuals’ motivation to receive vaccines against COVID-19.

The theory of planned behavior (TPB) [5,6,7,8] and health belief model (HBM) [9] are often used to understand the factors related to individuals’ decisions regarding vaccination. According to the TPB [5,6,7,8], individuals’ decisions regarding vaccination depend on multiple factors, including an assessment of the benefits and harms of vaccination (personal attitudes), perceived level of competence in deciding whether to undergo vaccination (self-control), perceived influence of significant others (social influences), and perceived dangers of the infectious disease against which the vaccine is intended to protect (risk assessment). According to the HBM [9], individuals’ beliefs in the consequences of contracting COVID-19, perceived benefits of and barriers to receiving vaccines against COVID-19, and self-efficacy explain the action to receive the COVID-19 vaccine [10]. The effects of these factors on the motivation to receive vaccination vary across different groups and must be considered individually.

Vaccination behaviors among individuals with schizophrenia needs to be investigated in depth for several reasons. First, individuals with schizophrenia are more likely to be infected with COVID-19 than are members of the general population [11]. Second, compared with the general population, individuals with schizophrenia have poorer prognoses after contracting COVID-19, including higher rates of morbidity, hospitalization, and mortality [11,12,13,14,15]. Third, according to the TPB [5,6,7,8] and HBM [9], individuals’ cognition and interactions with environments affect individuals’ motivation to receive the COVID-19 vaccine. However, both cognitive deficits [16] and social dysfunction [17] are core features of schizophrenia and may compromise the motivation to receive COVID-19 vaccines in individuals with schizophrenia. Therefore, vaccination against COVID-19 is especially crucial for individuals with schizophrenia.

Several studies have determined that individuals with severe mental illnesses have lower rates of COVID-19 vaccination than do the general population [18,19,20,21]. A longitudinal cohort study of 25,539 individuals with schizophrenia and 25,539 controls without schizophrenia from a healthcare database in Israel observed that those with schizophrenia had significantly higher rates of hospitalization and death due to COVID-19 and a significantly lower rate of COVID-19 vaccination than did the controls [14]. The same study continued to monitor booster vaccination for COVID-19 in both groups and determined that a higher proportion of the individuals with schizophrenia did not receive the booster vaccine than did in the control group, and if these patients did receive the booster, it was at a later time [22]. A study on 100 individuals with schizophrenia and 72 nonclinical controls in France found a lower rate of vaccination in individuals with schizophrenia compared to controls (64% versus 77.8%) [23]. A study on 112 individuals with schizophrenia in Tunisia found that 52.7% of participants refused to be vaccinated [24]. The aforementioned findings indicate that individuals with schizophrenia are at a disadvantage in terms of COVID-19 vaccination and require active assistance.

Previous cross-sectional studies have demonstrated that individual factors such as delusion and negative symptoms [25,26], low confidence in the positive effects of vaccines [19,27], concerns regarding the side effects of vaccines [19,26], and absence of comorbid physical illnesses [14] are associated with low motivation to receive vaccination against COVID-19 among individuals with schizophrenia. However, no prospective study has examined the predictions of several individual, environmental, and individual–environmental interaction factors for the motivation to receive the COVID-19 vaccine among individuals with schizophrenia. For example, according to the TPB [28,29], subjective behavioral control over vaccination and subjective norms of COVID-19 vaccination should be associated with the motivation to receive the COVID-19 vaccine [30]. Depression, self-esteem, and self-stigma associated with having a schizophrenia diagnosis may affect individuals’ self-efficacy in controlling their vaccination behaviors. In addition, limited social interaction may reduce individuals’ ability to understand vaccination norms. Increased perceived barriers to vaccination can predict low motivation to receive vaccination against COVID-19 [9,10]. Furthermore, depression, self-stigma, and lack of social support may increase the difficulties of individuals with schizophrenia in overcoming barriers to COVID-19 vaccination. An additional follow-up study is warranted to examine the predictive effects of individual, environmental, and individual–environmental interaction factors on the motivation to receive the COVID-19 vaccine and the number of COVID-19 vaccine doses received among individuals with schizophrenia.

The present 1-year follow-up study investigated the factors related to the motivation to receive the COVID-19 vaccine and the number of COVID-19 vaccine doses received among individuals with schizophrenia. We hypothesized that the level of motivation to receive vaccination would vary among individuals with schizophrenia with different sociodemographic and illness characteristics. In addition, we hypothesized that more severe depressive symptoms, self-stigma, and loneliness would predict lower motivation to receive vaccination and receipt of fewer vaccine doses and that greater self-esteem and social support would predict higher motivation to receive vaccination and receipt of more vaccine doses.

## 2. Methods

### 2.1. Participants and Procedure

The Study on Stigma and Social Relationships in Individuals with Schizophrenia [31,32] invited 300 individuals with the diagnosis of schizophrenia or schizoaffective disorder diagnosed by psychiatrists according to the fifth edition of the *Diagnostic and Statistical Manual of Mental Disorders* [33] from the psychiatric outpatient clinics of Kaohsiung Medical University Hospital and two institutes for psychiatric rehabilitation in the communities of Southern Taiwan from 1 February 2022 to 31 May 2022. The age distribution of the participants ranged from 20 to 70 years old (*Mage* = 44.6 years). To avoid participants not understanding the purpose of the study and the content of the questionnaire, individuals with the diagnosis of cognitive disorders due to physical problems (e.g., head injury, hepatic or renal diseases, and cerebrovascular diseases), intellectual disability, and alcohol and substance use disorder other than nicotine use disorder were excluded. The present study invited the same 300 individuals in the Study on Stigma and Social Relationships in Individuals with Schizophrenia to receive a follow-up interview 1 year later. All participants who participate in the follow-up study provided written informed consent. The institutional review board of Kaohsiung Medical University Hospital approved this study (approval number: KMUHIRB-SV(I)-20220063).

### 2.2. Measures

#### 2.2.1. Predicting Variables at Baseline

The participants’ individual factors (sociodemographic characteristics, psychiatric diagnoses and symptoms, and self-esteem), environmental factors (perceived social support from family and nonfamily members), and individual environmental factors (self-stigma and loneliness) were measured at baseline.

##### Sociodemographic Characteristics

Information on the participants’ gender, age, years of education completed, amount of money that can be freely spent in each month, current marital status, and occupational status was collected.

##### Psychiatric Diagnosis and Symptoms

Data regarding the participants’ psychiatric diagnosis (schizophrenia vs. schizoaffective disorder), duration of illness (the total number of years since the initial diagnosis), and psychiatric symptoms were collected. The present study used the Positive and Negative Syndrome Scale (PANSS) [34,35,36] to evaluate five domains of symptoms, including excitement, emotional distress, disorganization, positive symptoms, and negative symptoms. Each item was rated on a 7-point Likert scale, with a mean score representing the severity of symptoms. A higher mean score for each module indicates more severe psychiatric symptoms. Emotional distress was found to significantly relate to depression in this study; to prevent collinearity, emotional distress was not included in linear regression models. The present study also used the Center for Epidemiologic Studies Depression Scale (CES-D) [37,38] to evaluate participants’ frequency of depressive symptoms in the preceding month on a 4-point Likert scale, with a higher total score indicating more severe depression.

##### Global Self-Esteem

This study used the Rosenberg Self-Esteem Scale (RSES) [39] to measure the level of subjective global self-esteem. Each item is rated on a 4-point Likert scale, with a higher total score indicating a higher level of global self-esteem.

##### Self-Stigma

This study used the Self-Stigma Scale–Short (SSS-S) [40] to evaluate participants’ self-stigma regarding having a mental illness. Each item is rated on a 4-point Likert scale, with a higher total score indicating a higher level of self-stigma regarding having a mental illness.

##### Loneliness

This study used the University of California, Los Angeles (UCLA), Loneliness Scale (Version 3) [41,42] to evaluate participants’ levels of loneliness. Each item is rated on a 4-point Likert scale, with a higher total score indicating a higher level of loneliness.

##### Perceived Support from Family and Nonfamily Members

This study used the Family and Friend Adaptability, Partnership, Growth, Affection, and Resolve (APGAR) Index [43,44] to evaluate participants’ perceived support from family and nonfamily members, respectively. Each item was rated on a 4-point Likert scale; higher total scores indicate higher levels of perceived support from family and nonfamily members.

#### 2.2.2. Outcome Variables

##### Motivation to Receive Vaccination against COVID-19

This study used the 9-item version of the Motors of COVID-19 Vaccination Acceptance Scale (MoVac-COVID19S) [45,46] to evaluate participants’ motivation to receive vaccination against COVID-19. Each item is rated on a 7-point Likert scale, with a higher total score indicating greater motivation to receive vaccination against COVID-19.

##### Doses of Vaccine against COVID-19

The participants were asked “How many doses of vaccine against COVID-19 have you received thus far?”

### 2.3. Statistical Analysis

We performed statistical analyses using International Business Machines Corporation (IBM) SPSS Statistics version 24.0 (IBM Corporation, Armonk, NY, USA). We summarized and analyzed the participants’ sociodemographic characteristics (i.e., gender, age, educational level, occupation, and marital status), illness characteristics (diagnosis, duration of illness, and psychiatric symptoms on the PANSS), depressive symptoms, self-esteem, self-stigma due to having a mental illness, perceived family and friend support, loneliness, motivation to receive vaccination against COVID-19, and number of vaccine doses received by using descriptive statistics. We tested the extent of deviation from a normal distribution; the results did not reveal any severe deviation [47]. We performed bivariate linear regression analysis to examine the associations of the factors at baseline with motivation to receive vaccination against COVID-19 and the number of vaccine doses received at follow-up. Baseline factors that demonstrated a significant association with motivation to receive vaccination against COVID-19 and the number of vaccine doses received at follow-up in the bivariate linear regression were included in multivariate linear regression analysis. If collinearity was significant, stepwise multivariate linear regression analysis was performed. A *p* value of <0.05 was considered statistically significant.

## 3. Results

A total of 257 (85.7%) individuals participated in the follow-up study, 13 (4.3%) refused to participate in the follow-up study, and 30 (10%) were lost to follow-up. No differences in gender (χ^2^ = 2.814, *p* = 0.093), age (t = 0.863, *p* = 0.389), or number of years of education (t = 0.742, *p* = 0.459) were observed between those who completed the follow-up survey and those who did not.

Table 1 lists the sociodemographic and illness characteristics as well as the data on depression, self-esteem, self-stigma, perceived family and friend support, loneliness, motivation to receive vaccination against COVID-19, and number of vaccine doses received for the 257 participants who underwent the follow-up assessment (44.4% men and 55.6% women). Their mean age was 45.6 (standard deviation [*SD*] = 11.3) years at baseline. Their mean years of education was 13.0 (*SD* = years) years, and 86.4% of the participants were separated or divorced. Their mean monthly disposable income was TWD 7973.8 (*SD* = 8179.4). Furthermore, 70.4% of the participants were unemployed, and 87.5% were diagnosed as having schizophrenia. The mean duration of illness since the initial diagnosis was 19.0 (*SD* = 10.0) years. The mean scores for positive symptoms, negative symptoms, disorganization, and excitement on the PANSS were 3.5 (*SD* = 0.9), 3.6 (*SD* = 0.9), 3.5 (*SD* = 0.9), and 2.5 (*SD* = 0.9), respectively. The mean CES-D score was 16.7 (*SD* = 11.1). The mean RSES score was 28.1 (*SD* = 5.7). The mean SSS-S score was 20.0 (*SD* = 5.3). The mean UCLA Loneliness Scale score was 43.2 (*SD* = 11.5). The mean Family and Friend APGAR Index scores were 15.6 (*SD* = 3.7) and 13.5 (*SD* = 4.4), respectively. The mean MoVac-COVID19S score was 50.8 (*SD* = 9.5). The mean number of COVID-19 vaccine doses received was 3.3 (*SD* = 1.4).

Table 2 presents the findings for the associations of individual, environmental, and individual–environmental interaction factors with motivation to receive vaccination against COVID-19. The unadjusted models demonstrated that participants who were separated or divorced had greater motivation to receive vaccination than those who were married or cohabitating. Greater self-esteem (*p* < 0.05) and perceived support from family (*p* < 0.05) and nonfamily members (*p* < 0.05) at baseline were significantly associated with greater motivation to receive vaccination; by contrast, greater loneliness at baseline was significantly associated with lower motivation to receive vaccination (*p* < 0.01). These baseline factors that demonstrated a significant association with motivation to receive vaccination were included in a stepwise multiple linear regression analysis. The results indicated that being married or cohabitating (*p* < 0.05) and greater loneliness (*p* < 0.01) at baseline were significantly associated with lower motivation to receive vaccination against COVID-19.

Table 3 presents the findings for the associations of individual, environmental, and individual–environmental interaction factors with the number of COVID-19 vaccine doses. The unadjusted models demonstrated that greater negative symptoms (*p* < 0.05), disorganization (*p* < 0.01), and self-stigma (*p* < 0.05) at baseline were significantly associated with fewer COVID-19 vaccine doses received at follow-up; by contrast, greater motivation to receive vaccination was significantly associated with more doses (*p* < 0.001). These factors that demonstrated a significant association with the number of COVID-19 vaccine doses were included in a multiple linear regression analysis. The results indicated that disorganization (*p* < 0.05) at baseline was significantly associated with fewer COVID-19 vaccine doses at follow-up and that greater motivation to receive vaccination was significantly associated with more COVID-19 vaccine doses (*p* < 0.001).

## 4. Discussion

This follow-up study discovered that multiple individual, environmental, and individual–environmental interaction factors at baseline predicted motivation to receive vaccination against COVID-19 and the number of vaccine doses received at follow-up among individuals with schizophrenia, as demonstrated in bivariate linear regression analysis models. In multivariate linear regression analysis, being married or cohabitating and greater loneliness at baseline were associated with lower motivation to receive vaccination against COVID-19. Furthermore, greater disorganization at baseline and lower motivation to receive vaccination at follow-up were associated with fewer COVID-19 vaccine doses received at follow-up.

Greater loneliness at baseline predicted lower motivation to receive vaccination against COVID-19 1 year later, even after adjustment for other factors. Moreover, greater family and friend support at baseline predicted greater motivation to receive vaccination against COVID-19 1 year later in bivariate linear regression analysis. According to the TPB [28,29], social interaction can expose individuals to norms surrounding COVID-19 vaccination, and social support can enhance individuals’ self-efficacy in receiving vaccination. According to the HBM [9], social interaction can increase individuals’ knowledge of the benefits of COVID-19 vaccination, and social support can help individuals overcome barriers to vaccination. Loneliness refers to an individual’s subjective perceived distance between anticipated and actual levels of social connectivity [48]. Research found that loneliness increases the risks of pessimism [49], which may decrease individuals’ motivation to adopt self-protective behaviors such as receiving vaccination to against COVID-19. Moreover, loneliness increases the risk of substance abuse [50], which may increase individuals’ difficulty in obtaining the information and sources of vaccination. In addition to loneliness and social support, higher self-esteem at baseline predicted greater motivation to receive vaccination against COVID-19. Individuals with lower self-esteem have a lower sense of control over the decision to receive vaccination and will not be unable to make an informed decision regarding receiving vaccination. Furthermore, low self-esteem also increases the risk of loneliness [50] and further decreases the motivation to receive vaccination.

The present study found that individuals with schizophrenia who were separated or divorced had greater motivation to receive vaccination than did those who were married or cohabitating. The underlying reasons remain unexplored. It is possible that single people are more likely to receive the vaccine because of concerns about having no one to care for them in the event they contract COVID-19. It is also possible that societal pressures, post-separation mental status, and economic conditions contribute to the motivation to receive vaccination. Individuals who are divorced or separated might have experienced traumatic events or social upheavals that affected their mental health, which might further influence their vaccination decision making. Further study diving deeper into these nuances could provide more actionable insights. Greater motivation to receive vaccination was cross-sectionally associated with more COVID-19 vaccine doses among individuals with schizophrenia. The motivation to receive vaccination against COVID-19, as measured by the MoVac-COVID19S, encompasses multiple domains of cognition relating to vaccination, including the value of vaccination, its impact on health, knowledge about vaccination, and individual autonomy in the decision to receive vaccination [51,52]. All these aspects of vaccination-related cognition may affect individuals’ decision to receive vaccination. This study found that greater disorganization at baseline predicted fewer COVID-19 vaccine doses received at follow-up among the individuals with schizophrenia. Research has found that the disorganization factor of the PANSS, which includes difficulty in abstract thinking, poor attention, disorientation, stereotyped thinking, and conceptual disorganization, shows a significant association with cognitive test scores [53,54]. Therefore, disorganization in schizophrenia may not only increase individuals’ difficulties in completing the vaccination process, including scheduling appointments and attending them, but also cause these individuals to misjudge the pros and cons of receiving vaccination.

The present study found that greater negative symptoms at baseline predicted fewer COVID-19 vaccine doses received at follow-up among the individuals with schizophrenia. Negative symptoms are a core feature of schizophrenia that are predominant, enduring, and clinically relevant in up to 60% of patients [55]. Patients with prominent negative symptoms have worse functional outcomes, with observed correlations existing between negative symptoms and impaired occupational, household, and recreational functioning, as well as relationship difficulties [56,57]. Negative symptoms such as a lack of interest in the world and social withdrawal may reduce individuals’ opportunities to receive information regarding vaccination. Negative symptoms such as an inability to act spontaneously and lack of motivation may also inhibit individuals’ execution of vaccination registration and actual receipt of vaccination.

In addition, greater self-stigma at baseline predicted fewer COVID-19 vaccine doses received at follow-up among the individuals with schizophrenia. Self-stigma resulting from mental disorders refers to an individual’s concepts to accept and internalize negative public stereotypes about severe mental disorders [58,59]. Previous studies have indicated that self-stigma increases the risks of social anxiety, hopelessness, and low treatment adherence among individuals with mental illnesses [60,61,62]. Therefore, self-stigma may limit the ability of individuals with schizophrenia to complete procedures required for receiving vaccines. In addition to self-stigma, individuals with schizophrenia may experience societal stigma from the public [63], health professionals [64], and workplaces [65], as well as microaggression [66]. The effects of the multiple forms of stigmatizing experiences on the motivation to receive vaccination against COVID-19 among individuals with schizophrenia warrant further study.

### 4.1. Implications

We propose the following suggestions. Individuals with schizophrenia who experience social isolation and loneliness should be the focus of intervention programs aimed at increasing their motivation to receive vaccination against COVID-19. Strategies to enhance social support and reduce the feeling of loneliness can help increase the motivation to receive vaccination against COVID-19 among these individuals. Furthermore, enhancing individuals’ motivation to receive vaccination is essential for the COVID-19 vaccine. Individuals with schizophrenia who exhibit more severe negative symptoms, disorganization, and self-stigma may need special assistance to complete the vaccination process.

### 4.2. Strengths and Limitations

One strength of this study is its prospective design, which enabled us to examine temporal relationships between potential predictors and outcome variables. This study is the first to investigate the prediction of multiple individual, environmental, and individual–environmental interaction factors on the motivation to receive vaccination against COVID-19 among individuals with schizophrenia. However, this study has several limitations that should be addressed. First, because we collected data from a single source, our results may have been subject to shared-method variance. Moreover, this study relied on self-reporting measures, which might introduce recall biases, social desirability biases, and misinterpretation of questionnaire items. Future studies could benefit from triangulating self-reports with other data sources to enhance validity. Second, although this prospective study examined the associations between certain baseline factors and motivation to vaccinate or doses received at follow-up, causality cannot be inferred from these associations. There might be a potential confounding effect from other unaccounted variables, such as their access to healthcare, healthcare system trust, or societal attitudes towards those with schizophrenia in Taiwan. Third, whether our results can be generalized to individuals with schizophrenia who refused to participate in the follow-up study or those who did not seek medical help in outpatient clinics remains unclear. Whether these excluded individuals with schizophrenia might have a different level of the motivation to receive vaccination against COVID-19 and related factors warrants further study. Moreover, whether the results of this study can be generalized to the population living in areas of different cultural, socio-economic, and health infrastructure warrants further study. Fourth, we did not evaluate some baseline factors such as chronic physical illness and patients’ history of vaccination against influenza. Previous studies have found that comorbid physical illnesses [14] and regular vaccination against influenza [67] positively relate to the motivation of receiving the COVID-19 vaccination in individuals with schizophrenia. The predictive effects of these factors on the motivation to receive vaccination against COVID-19 remain undetermined. Fifth, the duration between the initial and follow-up study was one year in this study, which might not be long enough to capture dynamic shifts in attitudes and behaviors related to COVID-19 vaccination, especially given the rapidly evolving nature of the pandemic and related public health advice. Future studies are needed to consider more frequent follow-ups or a longer observational period.

## 5. Conclusions

The results of this follow-up study reveal that multiple individual, environmental, and individual–environmental interaction factors predict the motivation to receive vaccination against COVID-19 and the number of COVID-19 vaccine doses received among individuals with schizophrenia. However, the predictors of motivation differ from those of the number of vaccine doses received. Health professionals should consider the identified predictors while developing intervention programs aimed at enhancing COVID-19 vaccination among individuals with schizophrenia.

## Figures and Tables

**Table 1 vaccines-11-01781-t001:** Participant characteristics (*N* = 257).

Variable	*N* (%)	Mean (SD)	Range
Gender			
Female	143 (55.6)		
Male	114 (44.4)		
Age (year)		45.6 (11.3)	20–70
Education (year)		13.0 (2.7)	6–20
Marital status			
Separated or divorced	222 (86.4)		
Married or cohabited	35 (13.6)		
Monthly disposable income (TWD)		7973.8 (8179.4)	0–60,000
Occupation			
Unemployed	181 (70.4)		
Full-time or part-time job	76 (29.6)		
Diagnosis			
Schizophrenia	225 (87.5)		
Schizoaffective disorder	32 (12.5)		
Duration of illness (year)		19.0 (10.0)	0.50–43
Positive symptoms on the PANSS		3.5 (0.9)	2–6
Negative symptoms on the PANSS		3.6 (0.9)	1–6
Disorganization on the PANSS		3.5 (0.9)	1–6
Excitement on the PANSS		2.5 (0.9)	1–5
Depression		16.7 (11.1)	0–54
Self-esteem		28.1 (5.7)	11–40
Self-stigma		20.0 (5.3)	9–36
Loneliness		43.2 (11.5)	20–76
Perceived support from families		15.6 (3.7)	5–20
Perceived support from nonfamily members		13.5 (4.4)	5–20
Motivation to get vaccinated against COVID-19		50.8 (9.5)	15–63
Doses of vaccines against COVID-19		3.3 (1.4)	0–6

COVID-19: coronavirus disease 2019; PANSS: Positive and Negative Syndrome Scale.

**Table 2 vaccines-11-01781-t002:** Associations of individual, environmental, and individual–environmental interaction factors with motivation to receive vaccination against COVID-19: linear regression analysis.

	Unadjusted Model	Adjusted Model ^a^
	*B* (*se*)	*B* (*se*)
Gender ^b^	0.329 (1.194)	–
Age	–0.039 (0.052)	–
Education	–0.119 (0.224)	–
Monthly disposable income ^c^	–0.044 (0.073)	–
Marital status ^d^	–4.353 (1.708) *	–4.360 (1.681) *
Occupation ^e^	–1.851 (1.295)	–
Diagnosis ^f^	0.102 (1.797)	–
Duration of illness	0.020 (0.060)	–
Positive symptoms on the PANSS	0.497 (0.665)	–
Negative symptoms on the PANSS	–0.624 (0.628)	–
Disorganization on the PANSS	–0.815 (0.681)	–
Excitement on the PANSS	0.118 (0.079)	–
Depression	–0.077 (0.053)	–
Self-esteem	0.228 (0.103) *	–
Self-stigma	–0.168 (0.111)	–
Loneliness	–0.154 (0.051) **	–0.154 (0.050) **
Perceived support from families	0.347 (0.161) *	–
Perceived support from nonfamily members	0.277 (0.133) *	–

^a^: stepwise multiple linear regression; ^b^: female as the reference; ^c^: TWD 1000 per unit; ^d^: separated or divorced as the reference; ^e^: unemployed as the reference; ^f^: schizophrenia as the reference. * *p* < 0.05, ** *p* < 0.01. COVID-19: coronavirus disease 2019; PANSS: Positive and Negative Syndrome Scale.

**Table 3 vaccines-11-01781-t003:** Associations of individual, environmental, and individual–environmental interaction factors with the number of COVID-19 vaccine doses received: linear regression analysis.

	Unadjusted Model	Adjusted Model ^a^
	*B* (*se*)	*B* (*se*)
Gender ^b^	–0.039 (0.182)	–
Age (year)	–0.005 (0.008)	–
Education (year)	0.046 (0.034)	–
Money that could be spent freely ^c^	0.005 (0.011)	–
Marital status ^d^	–0.327 (0.263)	–
Occupation ^e^	0.137 (0.198)	–
Diagnosis ^f^	0.251 (0.274)	–
Duration of illness	0.005 (0.009)	–
Positive symptoms on the PANSS	–0.006 (0.102)	–
Negative symptoms on the PANSS	–0.218 (0.095) *	0.000 (0.114)
Disorganization on the PANSS	–0.345 (0.102) **	–0.283 (0.123) *
Excitement on the PANSS	0.013 (0.012)	–
Depression	–0.010 (0.008)	–
Self-esteem	0.017 (0.016)	–
Self-stigma	–0.034 (0.017) *	–0.01 (0.016)
Loneliness	–0.011 (0.008)	–
Perceived support from families	–0.006 (0.025)	–
Perceived support from nonfamily members	0.033 (0.020)	–
Motivation to get vaccinated against COVID-19	0.059 (0.009) ***	0.057 (0.009) ***

^a^: full-entered multiple linear regression; ^b^: female as the reference; ^c^: presented as TWD 1000 per unit; ^d^: separated or divorced as the reference; ^e^: unemployed as the reference; ^f^: schizophrenia as the reference. * *p* < 0.05; ** *p* < 0.01; *** *p* < 0.001. COVID-19: coronavirus disease 2019; PANSS: Positive and Negative Syndrome Scale.

## Data Availability

The data will be available upon reasonable request to the corresponding authors.

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
