# Peer review of "Predictors of Motivation to Receive a COVID-19 Vaccination and the Number of COVID-19 Vaccine Doses Received in Patients with Schizophrenia"

_vaccines, 2023, doi:10.3390/vaccines11121781_

Round 1

Reviewer 1 Report

Comments and Suggestions for Authors

Thanks for the opportunity to review this paper.

below are some suggestions;

1-Could you provide the specific sources for the COVID-19 statistics mentioned in the introduction, such as the number of cases and deaths on September 7, 2023?
2-How can you elaborate on the hesitancy to receive the COVID-19 vaccine?

3-What factors contribute to this hesitancy, and how widespread is it in different populations?

4-What research methods and data sources were used in the Israeli study on COVID-19 vaccination rates among individuals with schizophrenia?

5-Can you explain how ecological systems theory and the theory of planned behavior apply to understanding the motivation for COVID-19 vaccination among individuals with schizophrenia?

6-What are the key findings of this 1-year follow-up study, and how do they relate to the hypotheses presented in the introduction?

7-How does the study address the potential individual, environmental, and individual-environmental interaction factors that may influence motivation to receive the COVID-19 vaccine among individuals with schizophrenia?

8-Need to add more discussion around the results to strength the study.

9-Mention future perspective

Author Response

We appreciated your valuable comments. As discussed below, we have revised our manuscript with underlines based on your suggestions. Please let us know if we need to provide anything else regarding this revision.

Comment

1-Could you provide the specific sources for the COVID-19 statistics mentioned in the introduction, such as the number of cases and deaths on September 7, 2023?

Response

Thank you for your comment. This data come from WHO Coronavirus (COVID-19) Dashboard of World Health Organization. We provided the link to the source of the data in reference 1. We also updated the numbers of confirmed COVID-19 cases and death as below. Please refer to line 31-33.

On November 2, 2023, the World Health Organization (WHO) reported over 771,679,618 confirmed COVID-19 cases, including 6,977,023 deaths, worldwide [1].

Comment

2-How can you elaborate on the hesitancy to receive the COVID-19 vaccine?

Response

We added the definition of vaccine hesitancy as below into the revised manuscript. Please refer to line 35-36.

Vaccine hesitancy, which is defined by WHO as a “delay in acceptance or refusal of vaccines despite availability of vaccination services” [3].

Comment

3-What factors contribute to this hesitancy, and how widespread is it in different populations?

Response

We added the factors related to vaccine hesitancy among individuals with schizophrenia as below into the revised manuscript. Please refer to line 79-86.

Previous cross-sectional studies have demonstrated that individual factors such as delusion and negative symptoms [25,26], low confidence in the positive effects of vaccines [19,27], concerns regarding the side effects of vaccines [19,26], and absence of comorbid physical illnesses [14] are associated with low motivation to receive the COVID-19 vaccine among individuals with schizophrenia. However, no prospective study has examined the predictions of several individual, environmental, and individual–environmental interaction factors for the motivation to receive the COVID-19 vaccine among individuals with schizophrenia.

Comment

4-What research methods and data sources were used in the Israeli study on COVID-19 vaccination rates among individuals with schizophrenia?

Response

We added the research method and data source in the Israeli study as below into the revised manuscript. Please refer to line 65-67.

A longitudinal cohort study of 25,539 individuals with schizophrenia and 25,539 controls without schizophrenia from a healthcare database in Israel...

Comment

5-Can you explain how ecological systems theory and the theory of planned behavior apply to understanding the motivation for COVID-19 vaccination among individuals with schizophrenia?

Response

We added the core components of the theory of planned behavior and health belief model as below into the revised manuscript. To simplify the theoretical frame, we deleted the content of the ecological systems theory from the revised manuscript.

The theory of planned behavior (TPB) [5–8] and health belief model (HBM) [9] are often used to understand the factors related to individuals’ decisions regarding vaccination. According to the TPB [5–8], individuals’ decisions regarding vaccination depend on multiple factors, including an assessment of the benefits and harms of vaccination (personal attitudes), perceived level of competence in deciding whether to undergo vaccination (self-control), perceived influence of significant others (social influences), and perceived dangers of the infectious disease against which the vaccine is intended to protect (risk assessment). According to the HBM [9], individuals’ beliefs in the consequences of contracting COVID-19, perceived benefits of and barriers to receiving vaccines against COVID-19, and self-efficacy explain the action to receive the COVID-19 vaccine [10].” Please refer to line 41-50.

“…according to the TPB [5–8] and HBM [9], individuals’ cognition and interactions with environments affect individuals’ motivation to receive the COVID-19 vaccine. However, both cognitive deficits [16] and social dysfunction [17] are core features of schizophrenia and may compromise the motivation to receive COVID-19 vaccines in individuals with schizophrenia. Therefore, vaccination against COVID-19 is especially crucial for individuals with schizophrenia. Please refer to line 58-63.

Comment

6-What are the key findings of this 1-year follow-up study, and how do they relate to the hypotheses presented in the introduction?

Response

We described the hypotheses and key findings of this study as below into the revised manuscript.

Hypotheses: “We hypothesized that the level of motivation to receive vaccination would vary among individuals with schizophrenia with different sociodemographic and illness characteristics. In addition, we hypothesized that more severe depressive symptoms, self-stigma, and loneliness would predict lower motivation to receive vaccination and receipt of fewer vaccine doses and that greater self-esteem and social support would predict higher motivation to receive vaccination and receipt of more vaccine doses.” Please refer to line 101-107.

Key findings: “This follow-up study discovered that multiple individual, environmental, and individual–environmental interaction factors at baseline predicted motivation to receive COVID-19 vaccination and the number of vaccine doses received at follow-up among individuals with schizophrenia, as demonstrated in bivariate linear regression analysis models. In multivariate linear regression analysis, being married or cohabitating and greater loneliness at baseline were associated with lower motivation to receive COVID-19 vaccination. Furthermore, greater disorganization at baseline and lower motivation to receive vaccination at follow-up were associated with fewer COVID-19 vaccine doses received at follow-up.” Please refer to line 267-275.

Comment

7-How does the study address the potential individual, environmental, and individual-environmental interaction factors that may influence motivation to receive the COVID-19 vaccine among individuals with schizophrenia?

Response

Thank you for your comment. In this study we investigated the effects of individual (sociodemographic and illness characteristics, depression, and self-esteem), environmental (perceived social support), and individual–environmental interaction factors (self-stigma and loneliness) on the motivation to receive the COVID-19 vaccine and the number of COVID-19 vaccine doses received among individuals with schizophrenia. We found that multiple individual, environmental, and individual–environmental interaction factors at baseline predicted motivation to receive COVID-19 vaccination and the number of vaccine doses received at follow-up among individuals with schizophrenia (please refer to line 267-270). On the basis of our study findings, we also propose suggestions for developing intervention strategies regarding the roles of individual, environmental, and individual–environmental interaction factors (please refer to line 343-351).

Comment

8-Need to add more discussion around the results to strength the study.

Response

Thank you for your comment. We rewrote the contents of Discussion to strengthen the value of this study. Please refer to line 266-386.

Comment

9-Mention future perspective

Response

We added future perspectives as below.

“The underlying reasons remain unexplored. It is possible that single people are more likely to receive the vaccine because of concerns about having no one to care for them in the event they contract COVID-19. It is also possible that societal pressures, post-separation mental status and economic conditions contribute to the motivation to receive vaccination. Individuals who are divorced or separated might have experienced traumatic events or social upheavals that affected their mental health, which might further influence their vaccination decision-making. Further study diving deeper into these nuances could provide more actionable insights.Please refer to line 397-304.

“In addition to self-stigma, individuals with schizophrenia may experience societal stigma from the public [63], health professionals [64], and workplaces [65], as well as microaggression [66]. The effects of the multiple forms of stigmatizing experiences on the motivation to receive vaccination against COVID-19 among individuals with schizophrenia warrant further study.Please refer to line 337-341.

“This study excluded individuals with any intellectual disability, substance use disorder other than nicotine use disorder, or cognitive dysfunction resulting from severe physical diseases. Whether these excluded individuals with schizophrenia might have a different level of the motivation to receive vaccination against COVID-19 and related factors warrants further study.Please refer to line 369-373.

“Whether the results of this study can be generalized to the population living in the areas of different cultural, socio-economic, and health infrastructure warrants further study.Please refer to line 374-376.

“The duration between the initial and follow-up study was one year in this study, which might not be long enough to capture dynamic shifts in attitudes and behaviors related to COVID-19 vaccination, especially given the rapidly evolving nature of the pandemic and related public health advice. Future study is needed to consider more frequent follow-ups or a longer observational period.” Please refer to line 381-386.

Reviewer 2 Report

Comments and Suggestions for Authors

The paper titled "Predictors of Motivation to Receive COVID-19 Vaccination and Number of COVID-19 Vaccine Doses Received in Patients with Schizophrenia" appears intriguing. However, it does have some significant shortcomings that the authors should address. My comments are provided in numbered form below.

1. The abstract seems to overemphasize certain elements while underemphasizing others. For instance, the details about the linear regression analysis could be condensed, and more focus could be given to the actual implications of the study's findings. It's essential to strike a balance between providing methodological details and underscoring the research's real-world relevance.

2. The introduction presents a fragmented overview of multiple theories and models (TPB, ecological systems theory, HBM) without seamlessly integrating them. It might be more effective to first present the general context of the problem, followed by a deeper dive into relevant theories that the study intends to employ or test, ensuring smoother transitions between the concepts.

3. While the introduction highlights the importance of understanding vaccination hesitancy in individuals with schizophrenia, it could be enhanced by providing a clearer, consolidated statement on why this specific group's vaccination behaviors are crucial to investigate, beyond just their heightened risk.

4. The mention of the Israeli study provides relevant context, but the level of detail about this specific study seems disproportionate compared to other presented facts. It could be more concise or balanced with other equally significant studies to prevent overshadowing other points.

5. The introduction's latter part discusses the present study's aims, hypotheses, and anticipated findings. While it's essential to lay out the study's objectives, the repeated mention of individual, environmental, and individual–environmental factors become unnecessary. Streamlining this section by avoiding repetitive terminology might improve readability.

6. The introduction could benefit from a clearer articulation of the study's novelty. While it's implied that previous research has not delved into multi-dimensional factors, explicitly stating what sets this study apart from existing literature can help highlight its significance. 

7. The study heavily relies on self-reporting measures, which can introduce a range of biases, including recall bias, social desirability bias, and misinterpretation of questionnaire items. Future studies could benefit from utilizing more objective assessments or triangulating self-report with other data sources to enhance validity.

8. The exclusion criteria regarding intellectual disability and cognitive dysfunction due to physical ailments might introduce selection bias, excluding a subset of the population that might have distinct perceptions and motivations about vaccination. To provide a comprehensive understanding, it's crucial to consider these excluded groups or at least discuss the potential implications of their exclusion.

9. The duration between the initial and follow-up study is one year, which might not be long enough to capture dynamic shifts in attitudes and behaviors related to COVID-19 vaccination, especially given the rapidly evolving nature of the pandemic and related public health advice. Future studies could consider more frequent follow-ups or a longer observational period.

10. The study was conducted in a specific region (Kaohsiung, Taiwan), which could limit the generalizability of the findings. Considering cultural, socio-economic, and health infrastructure differences across regions, it would be essential to replicate this study in various settings or at least acknowledge this limitation and discuss its potential impact on the findings.

11. The study provides a broad range of sociodemographic and illness characteristics but does not elucidate how these variables, particularly the illness characteristics, might biologically or psychologically influence one's motivation to get vaccinated. There needs to be a deeper exploration or theorization to explain the relationship, especially when specific factors such as disorganization and negative symptoms seem to impact vaccination doses.

12. The results indicate significant associations between certain baseline factors and motivation to vaccinate or doses received. However, causality cannot be inferred from these associations. Given the cohort studied, there's a potential confounding effect from other unaccounted variables, such as their access to healthcare, healthcare system trust, or societal attitudes towards those with schizophrenia in Taiwan.

13. While the results highlight a differential motivation between separated/divorced individuals and those married/cohabitating, the underlying reasons remain unexplored. This difference could be attributed to numerous external variables – societal pressures, individual psychology post-separation, or even economic conditions post-separation. Diving deeper into these nuances could provide more actionable insights.

14. The authors make strong connections between loneliness and lack of motivation for vaccination, but the underlying reasons behind this loneliness – particularly among schizophrenic individuals – are not fully explored. They attribute this to a lack of social exposure to vaccination norms and benefits, but a deeper understanding of how schizophrenia affects loneliness and how that, in turn, affects vaccine perceptions would strengthen the discussion.

15. The paper establishes correlations between certain baseline factors (e.g., self-esteem, marital status) and vaccination motivation but does not explore deeper potential confounders. For instance, individuals who are divorced or separated might have experienced traumatic events or social upheavals that affected their mental health, which might further influence their vaccination decision-making. This depth is missing in the current discussion.

16. While the paper addresses self-stigma as a limiting factor for vaccination uptake, it misses an opportunity to discuss broader societal stigma towards individuals with schizophrenia and how that might influence their motivation to get vaccinated, especially in the context of public health crises like the COVID-19 pandemic.

17. The discussion, while providing strengths and limitations, does not elaborate on potential improvements or modifications to the study design. The authors acknowledge shared-method variance and potential social desirability bias but do not provide suggestions on how future studies might mitigate these concerns. Additionally, while they mention unassessed factors like chronic illnesses and vaccination history, they don't delve into how these factors could have fundamentally changed their findings or discussion.

I hope that the provided comments serve to further strengthen the rigor and depth of this paper. They are intended to highlight areas of potential enhancement and are not a critique of the research's foundational merit. It is my sincere hope that the authors perceive these insights in the constructive spirit with which they are shared. Thank you for the opportunity to review this significant piece of work.

Comments on the Quality of English Language

Fine

Author Response

We appreciated your valuable comments. As discussed below, we have revised our manuscript with underlines based on your suggestions. Please let us know if we need to provide anything else regarding this revision.

Comment

  1. The abstract seems to overemphasize certain elements while underemphasizing others. For instance, the details about the linear regression analysis could be condensed, and more focus could be given to the actual implications of the study's findings. It's essential to strike a balance between providing methodological details and underscoring the research's real-world relevance.

Response

Thank you for your comment. We revised the abstract and condensed the content regarding methodology as below. Please refer to line 14-20.

“…The present 1-year follow-up study investigated the effects of individual (sociodemographic and illness characteristics, depression, and self-esteem), environmental (perceived social support), and individual–environmental interaction factors (self-stigma and loneliness) on the motivation to receive the COVID-19 vaccine and the number of COVID-19 vaccine doses received among 300 individuals with schizophrenia. The associations of baseline factors with motivation to receive COVID-19 vaccination and the number of vaccine doses received 1 year later were examined through linear regression analysis….

Comment

  1. The introduction presents a fragmented overview of multiple theories and models (TPB, ecological systems theory, HBM) without seamlessly integrating them. It might be more effective to first present the general context of the problem, followed by a deeper dive into relevant theories that the study intends to employ or test, ensuring smoother transitions between the concepts.

Response

Thank you for your comment. We revised the content of Introduction regarding the theories applied in this study as below to ensure smoother transitions between the concepts.

The theory of planned behavior (TPB) [5–8] and health belief model (HBM) [9] are often used to understand the factors related to individuals’ decisions regarding vaccination. According to the TPB [5–8], individuals’ decisions regarding vaccination depend on multiple factors, including an assessment of the benefits and harms of vaccination (personal attitudes), perceived level of competence in deciding whether to undergo vaccination (self-control), perceived influence of significant others (social influences), and perceived dangers of the infectious disease against which the vaccine is intended to protect (risk assessment). According to the HBM [9], individuals’ beliefs in the consequences of contracting COVID-19, perceived benefits of and barriers to receiving vaccines against COVID-19, and self-efficacy explain the action to receive the COVID-19 vaccine [10].” Please refer to line 41-50.

“…according to the TPB [5–8] and HBM [9], individuals’ cognition and interactions with environments affect individuals’ motivation to receive the COVID-19 vaccine. However, both cognitive deficits [16] and social dysfunction [17] are core features of schizophrenia and may compromise the motivation to receive COVID-19 vaccines in individuals with schizophrenia. Therefore, vaccination against COVID-19 is especially crucial for individuals with schizophrenia. Please refer to line 58-63.

Comment

  1. While the introduction highlights the importance of understanding vaccination hesitancy in individuals with schizophrenia, it could be enhanced by providing a clearer, consolidated statement on why this specific group's vaccination behaviors are crucial to investigate, beyond just their heightened risk.

Response

We rewrote the statement on why vaccination behaviors in individuals with schizophrenia are crucial to investigate as below. Please refer to line 53-63.

Vaccination behaviors among individuals with schizophrenia needs to be investigated in depth for several reasons. First, individuals with schizophrenia are more likely to be infected with COVID-19 than are members of the general population [11]. Second, compared with the general population, individuals with schizophrenia have poorer prognoses after contracting COVID-19, including higher rates of morbidity, hospitalization, and mortality [11–15]. Third, according to the TPB [5–8] and HBM [9], individuals’ cognition and interactions with environments affect individuals’ motivation to receive the COVID-19 vaccine. However, both cognitive deficits [16] and social dysfunction [17] are core features of schizophrenia and may compromise the motivation to receive COVID-19 vaccines in individuals with schizophrenia. Therefore, vaccination against COVID-19 is especially crucial for individuals with schizophrenia.

Comment

  1. The mention of the Israeli study provides relevant context, but the level of detail about this specific study seems disproportionate compared to other presented facts. It could be more concise or balanced with other equally significant studies to prevent overshadowing other points.

Response

Thank you for your comment. We added more results of studies on vaccination against COVID-19 in individuals with schizophrenia as below. Please refer to line 73-78.

A study on 100 individuals with schizophrenia and 72 nonclinical controls in France found a lower rate of vaccination in individuals with schizophrenia compared to controls (64% versus 77.8%) [23]. A study on 112 individuals with schizophrenia in Tunisia found that 52.7% of participants refused to be vaccinated [24]. The aforementioned findings indicate that individuals with schizophrenia are at a disadvantage in terms of COVID-19 vaccination and require active assistance.

Comment

  1. The introduction's latter part discusses the present study's aims, hypotheses, and anticipated findings. While it's essential to lay out the study's objectives, the repeated mention of individual, environmental, and individual–environmental factors become unnecessary. Streamlining this section by avoiding repetitive terminology might improve readability.

Response

We rewrote the last paragraph of Discussion describing the aims and hypotheses to avoid repetitive terminology as below. Please refer to line 99-107.

The present 1-year follow-up study investigated the factors related to the motivation to receive the COVID-19 vaccine and the number of COVID-19 vaccine doses received among individuals with schizophrenia. We hypothesized that the level of motivation to receive vaccination would vary among individuals with schizophrenia with different sociodemographic and illness characteristics. In addition, we hypothesized that more severe depressive symptoms, self-stigma, and loneliness would predict lower motivation to receive vaccination and receipt of fewer vaccine doses and that greater self-esteem and social support would predict higher motivation to receive vaccination and receipt of more vaccine doses.

Comment

  1. The introduction could benefit from a clearer articulation of the study's novelty. While it's implied that previous research has not delved into multi-dimensional factors, explicitly stating what sets this study apart from existing literature can help highlight its significance. 

Response

We added a clearer articulation of the study's novelty as below. Please refer to line 79-86.

Previous cross-sectional studies have demonstrated that individual factors such as delusion and negative symptoms [25,26], low confidence in the positive effects of vaccines [19,27], concerns regarding the side effects of vaccines [19,26], and absence of comorbid physical illnesses [14] are associated with low motivation to receive the COVID-19 vaccine among individuals with schizophrenia. However, no prospective study has examined the predictions of several individual, environmental, and individual–environmental interaction factors for the motivation to receive the COVID-19 vaccine among individuals with schizophrenia.

Comment

  1. The study heavily relies on self-reporting measures, which can introduce a range of biases, including recall bias, social desirability bias, and misinterpretation of questionnaire items. Future studies could benefit from utilizing more objective assessments or triangulating self-report with other data sources to enhance validity.

Response

Thank you for your comment. We added it as one of the limitations of this study as below. Please refer to line 359-362.

“…this study relied on self-reporting measures, which might introduce recall biases, social desirability biases, and misinterpretation of questionnaire items. Future studies could benefit from triangulating self-report with other data sources to enhance validity.

Comment

  1. The exclusion criteria regarding intellectual disability and cognitive dysfunction due to physical ailments might introduce selection bias, excluding a subset of the population that might have distinct perceptions and motivations about vaccination. To provide a comprehensive understanding, it's crucial to consider these excluded groups or at least discuss the potential implications of their exclusion.

Response

We added it as one of the issues warranted further study. Please refer to line 36-374.

Especially, this study excluded individuals with any intellectual disability, substance use disorder other than nicotine use disorder, or cognitive dysfunction resulting from severe physical diseases. Whether these excluded individuals with schizophrenia might have a different level of the motivation to receive vaccination against COVID-19 and related factors warrants further study.

Comment

  1. The duration between the initial and follow-up study is one year, which might not be long enough to capture dynamic shifts in attitudes and behaviors related to COVID-19 vaccination, especially given the rapidly evolving nature of the pandemic and related public health advice. Future studies could consider more frequent follow-ups or a longer observational period.

Response

Thank you for your comment. We also added it as one of the issues warranted further study. Please refer to line 381-386.

“…the duration between the initial and follow-up study was one year in this study, which might not be long enough to capture dynamic shifts in attitudes and behaviors related to COVID-19 vaccination, especially given the rapidly evolving nature of the pandemic and related public health advice. Future study is needed to consider more frequent follow-ups or a longer observational period.

Comment

  1. The study was conducted in a specific region (Kaohsiung, Taiwan), which could limit the generalizability of the findings. Considering cultural, socio-economic, and health infrastructure differences across regions, it would be essential to replicate this study in various settings or at least acknowledge this limitation and discuss its potential impact on the findings.

Response

We agreed the generalization might be limited due to the differences in cultural, socio-economic, and health infrastructure across regions. We added it as one of the limitations of this study as below. Please refer to line 374-376.

“…whether the results of this study can be generalized to the population living in the areas of different cultural, socio-economic, and health infrastructure warrants further study.

Comment

  1. The study provides a broad range of sociodemographic and illness characteristics but does not elucidate how these variables, particularly the illness characteristics, might biologically or psychologically influence one's motivation to get vaccinated. There needs to be a deeper exploration or theorization to explain the relationship, especially when specific factors such as disorganization and negative symptoms seem to impact vaccination doses.

Response

Thank you for your comment. We added more discussion regarding the roles of disorganization and negative symptoms for vaccination among individuals with schizophrenia as below. Please refer to line 310-329.

This study found that greater disorganization at baseline predicted fewer COVID-19 vaccine doses received at follow-up among the individuals with schizophrenia. Research has found that the disorganization factor of the PANSS which includes difficulty in abstract thinking, poor attention, disorientation, stereotyped thinking and conceptual disorganization shows a significant association with cognitive test scores [53,54]. Therefore, disorganization in schizophrenia may not only increase individuals’ difficulties in completing the vaccination process, including scheduling appointments and attending them, but also misjudge the pros and cons of receiving vaccination.

The present study found that greater negative symptoms at baseline predicted fewer COVID-19 vaccine doses received at follow-up among the individuals with schizophrenia. Negative symptoms are a core feature of schizophrenia that are predominant, enduring and clinically relevant in up to 60 % of patients [55]. Patients with prominent negative symptoms have worse functional outcomes, with observed correlations existing between negative symptoms and impaired occupational, household, and recreational functioning, as well as relationship difficulties [56,57]. Negative symptoms such as a lack of interest in the world and social withdrawal may reduce individuals’ opportunities to receive information regarding vaccination. Negative symptoms such as an inability to act spontaneously and lack of motivation may also inhibit individuals’ execution of vaccination registration and actual receipt of vaccination.

Comment

  1. The results indicate significant associations between certain baseline factors and motivation to vaccinate or doses received. However, causality cannot be inferred from these associations. Given the cohort studied, there's a potential confounding effect from other unaccounted variables, such as their access to healthcare, healthcare system trust, or societal attitudes towards those with schizophrenia in Taiwan.

Response

Thank you for your reminding. We added it as one of the limitations of this study as below. Please refer to line 362-367.

“…although this prospective study examined the associations between certain baseline factors and motivation to vaccinate or doses received at follow-up, causality cannot be inferred from these associations. There might be a potential confounding effect from other unaccounted variables, such as their access to healthcare, healthcare system trust, or societal attitudes towards those with schizophrenia in Taiwan.

Comment

  1. While the results highlight a differential motivation between separated/divorced individuals and those married/cohabitating, the underlying reasons remain unexplored. This difference could be attributed to numerous external variables – societal pressures, individual psychology post-separation, or even economic conditions post-separation. Diving deeper into these nuances could provide more actionable insights.

Response

Thank you for your comment. We added it into Discussion as below. Please refer to line 397-304.

The underlying reasons remain unexplored. It is possible that single people are more likely to receive the vaccine because of concerns about having no one to care for them in the event they contract COVID-19. It is also possible that societal pressures, post-separation mental status and economic conditions contribute to the motivation to receive vaccination. Individuals who are divorced or separated might have experienced traumatic events or social upheavals that affected their mental health, which might further influence their vaccination decision-making. Further study diving deeper into these nuances could provide more actionable insights.

Comment

  1. The authors make strong connections between loneliness and lack of motivation for vaccination, but the underlying reasons behind this loneliness – particularly among schizophrenic individuals – are not fully explored. They attribute this to a lack of social exposure to vaccination norms and benefits, but a deeper understanding of how schizophrenia affects loneliness and how that, in turn, affects vaccine perceptions would strengthen the discussion.

Response

Thank you for your comment. Please refer to line 284-289 and 293-294.

Loneliness refers to an individual’s perceived distance between anticipated and actual levels of social connectivity [48]. Research found that loneliness increases the risks of pessimism [49], which may decrease individuals’ motivation to adopt self-protective behaviors such as receiving vaccination to against COVID-19. Moreover, loneliness increases the risk of substance abuse [50], which may increase individuals’ difficulty in obtaining the information and sources of vaccination. Furthermore, low self-esteem also increases the risk of loneliness [50] and further decreases the motivation to receive vaccination.

Comment

  1. The paper establishes correlations between certain baseline factors (e.g., self-esteem, marital status) and vaccination motivation but does not explore deeper potential confounders. For instance, individuals who are divorced or separated might have experienced traumatic events or social upheavals that affected their mental health, which might further influence their vaccination decision-making. This depth is missing in the current discussion.

Response

We added more discussion regarding this issue as below. Please refer to line 397-304.

The underlying reasons remain unexplored. It is possible that single people are more likely to receive the vaccine because of concerns about having no one to care for them in the event they contract COVID-19. It is also possible that societal pressures, post-separation mental status and economic conditions contribute to the motivation to receive vaccination. Individuals who are divorced or separated might have experienced traumatic events or social upheavals that affected their mental health, which might further influence their vaccination decision-making. Further study diving deeper into these nuances could provide more actionable insights.

Comment

  1. While the paper addresses self-stigma as a limiting factor for vaccination uptake, it misses an opportunity to discuss broader societal stigma towards individuals with schizophrenia and how that might influence their motivation to get vaccinated, especially in the context of public health crises like the COVID-19 pandemic.

Response

Thank you for your comment. We added discussion regarding the role of societal stigma as below. Please refer to line 337-341.

In addition to self-stigma, individuals with schizophrenia may experience societal stigma from the public [63], health professionals [64], and workplaces [65], as well as microaggression [66]. The effects of the multiple forms of stigmatizing experiences on the motivation to receive vaccination against COVID-19 among individuals with schizophrenia warrant further study.

Comment

  1. The discussion, while providing strengths and limitations, does not elaborate on potential improvements or modifications to the study design. The authors acknowledge shared-method variance and potential social desirability bias but do not provide suggestions on how future studies might mitigate these concerns. Additionally, while they mention unassessed factors like chronic illnesses and vaccination history, they don't delve into how these factors could have fundamentally changed their findings or discussion.

Response

Thank you for your comment. We added more discussion about future study with the improvement of study design as below. Please refer to line XXX.

“The underlying reasons remain unexplored. It is possible that single people are more likely to receive the vaccine because of concerns about having no one to care for them in the event they contract COVID-19. It is also possible that societal pressures, post-separation mental status and economic conditions contribute to the motivation to receive vaccination. Individuals who are divorced or separated might have experienced traumatic events or social upheavals that affected their mental health, which might further influence their vaccination decision-making. Further study diving deeper into these nuances could provide more actionable insights.Please refer to line 397-304.

“In addition to self-stigma, individuals with schizophrenia may experience societal stigma from the public [63], health professionals [64], and workplaces [65], as well as microaggression [66]. The effects of the multiple forms of stigmatizing experiences on the motivation to receive vaccination against COVID-19 among individuals with schizophrenia warrant further study.Please refer to line 337-341.

“This study excluded individuals with any intellectual disability, substance use disorder other than nicotine use disorder, or cognitive dysfunction resulting from severe physical diseases. Whether these excluded individuals with schizophrenia might have a different level of the motivation to receive vaccination against COVID-19 and related factors warrants further study.Please refer to line 369-373.

“Whether the results of this study can be generalized to the population living in the areas of different cultural, socio-economic, and health infrastructure warrants further study.Please refer to line 374-376.

Previous studies have found that comorbid physical illnesses [14] and regular vaccination against influenza [67] are associated with high motivation to receive the COVID-19 vaccine among individuals with schizophrenia. The predictive effects of these factors on the motivation to receive COVID-19 vaccination remain undetermined. Please refer to line 377-381.

“The duration between the initial and follow-up study was one year in this study, which might not be long enough to capture dynamic shifts in attitudes and behaviors related to COVID-19 vaccination, especially given the rapidly evolving nature of the pandemic and related public health advice. Future study is needed to consider more frequent follow-ups or a longer observational period.Please refer to line 381-386.

Round 2

Reviewer 1 Report

Comments and Suggestions for Authors

I agree with the revisions, and the paper can be accepted now.

Reviewer 2 Report

Comments and Suggestions for Authors

Accepted.

Comments on the Quality of English Language

Need improvement